# Identification of Body Balance Deterioration of Gait in Women Using Accelerometers

**Raquel Leirós-Rodríguez [1],\*, Vicente Romo-Pérez [2], Jose L. García-Soidán [2]**  **and Anxela Soto-Rodríguez [3]**

[1]  Faculty of Physical Therapy, Universidade de Vigo, Campus A Xunqueira s/n, 36005 Pontevedra, Spain
[2]  Faculty of Sports Sciences and Education, Universidade de Vigo, Campus A Xunqueira s/n, 36005 Pontevedra, Spain; vicente@uvigo.es (V.R.-P.); jlsoidan@uvigo.es (J.L.G.-S.)
[3]  Health Service from Galicia (SERGAS), Galician Health Services—Ourense Hospital, 32005 Ourense, Spain; anxelasoro@hotmail.com
\*  Correspondence: rleiros@uvigo.es

**Abstract:** This study presents a simple methodology for the evaluation of gait with accelerometers, for rapid and simple application, in which we employ current balance tests in clinical practice (Timed Up and Go, Chair Stand Test and Six-Minute Walk Test). The aim was to determine whether the accelerometric valuation of gait can detect alterations in balance. The sample of this cross-sectional research, made during the months of May and June 2018, was composed of 145 healthy adult women (x = 63.8 ± 8.41 years), from the city of Ourense (Spain). They walked with a triaxial accelerometer at the fourth lumbar vertebra for a distance of 20 m. The test was repeated three times, and the mean of the three measurements was used in the analysis. There was a reduction in the values of acceleration recorded along all three of the axes and in the root mean square as the age increased. This reduction was very significant for the minimum values registered along the vertical and transverse axes, and for the maximum values along the mediolateral axis. Only the maximum values of the vector module demonstrated significant differences among the three age groups. A regression model allowed us to identify the values that give more information on the Timed Up and Go Test, namely: the maximum values of the root mean square and the mediolateral axis. An exhaustive analysis of the vertical and mediolateral axes and the vector module allows for the detection of early alterations in the automatic gait pattern.

**Keywords:** age; anthropometry; gait; falls; functional performance

## 1. Introduction

Every year, an estimated 30%–40% of patients over the age of 65 will fall at least once, and approximately half of those who fall do so repeatedly [1]. An early diagnosis of the deterioration of balance allows for a reduction in the number of falls among the elderly. These are a direct source of morbidity and mortality from. Indirectly, there are also important psychosocial consequences, including the fear of falling and self-isolation imposed on elders after a fall [1,2]. Therefore, a reduction in the incidence of falls and the injuries resulting from them can result in a major decrease in costs to the healthcare system [2]. The identification of factors that impair gait stability is critical to designing interventions that maintain seniors' independence and mobility. This is especially important in women, because of their longer life expectancy and greater incidence of falls in relation to men [3].

Research studies have mostly based their results on analyses conducted with force platforms and electronic walkways [4,5]. These tools provide results based on the behaviour of the center of pressure of the body (based on ground reaction forces and spatiotemporal gait parameters). This parameter

has been linked to a risk of falling, but it is not a reflection of the overall performance of the body in space [6,7]. In contrast, in a clinical setting, the use of tests such as the Romberg test, the Timed Up and Go, the Berg Balance Score, the Tinetti Scale, the Functional Reach Test or the BESTest (Balance Evaluation Systems Test) is frequent. In general, one aspect common to all of them is that the resulting score depends on the qualitative and subjective evaluation of the evaluator [8].

An alternative, low-cost and portable method that is easy to apply to the analysis of the kinematic movements of an individual is the use of accelerometers [1]. These can quantify the movements of any body segment and can be used for the study of equilibrium. Accelerometers are used to evaluate center-of-mass dynamic behaviour [9,10]. Previous studies have shown the sensitivity of these devices to small changes in postural control systems [10–12]. Gait analysis, based on the study of the acceleration of the body, has been a valid and reliable method for predicting the risk of falling or discerning population subgroups [13–15]. The study of body kinematics facilitates the early detection of alterations in the gait when they are not yet detectable through visual analysis [16].

The use of accelerometers in research allows for identifying gait characteristics that provide additional information about the patient's degree of functionality or risk of suffering a fall. In addition, it is a more objective alternative than the use of clinical assessment scales [17–19].

Therefore, this work presents a methodology for the evaluation of gait with accelerometers for quick and easy application, in which we employ currently used balance tests in clinical practice. Although the accelerometric results have previously already been compared with clinical evaluation tests, this study includes the broadest sample and includes the clinical tests most used by physicians. The aim of this research is to determine whether the accelerometric valuation of gait can detect alterations in balance during aging (through comparing different age groups), with the hypothesis that trunk accelerations during gait vary with aging and that accelerometers are able to identify such changes.

## 2. Materials and Methods

### 2.1. Participants

The sample was composed of 145 adult healthy women from the city of Ourense (Spain), with an average age of 63.8 ± 8.4 years and an age range between 38 and 86 years. All of them were recruited from four municipal sports centers during the months of May and June 2018. At the time of recruitment, 523 women regularly attended these municipal sports centers. Of these, 281 women (53.7%) were eligible to participate in the study and agreed to be evaluated. The city of Ourense had a population of 160,867 women in 2018. To reach a confidence level of 95% and a margin of error of 6.9%, a minimum participation of 96 women was established. The following inclusion criteria were used: (a) engaged in physical activity between one and two days/week, and (b) walking between 30 and 90 min four days per week.

The exclusion criteria were as follows: (a) the inability to walk independently, (b) the use of external orthopaedic elements to maintain bipodal static balance with eyes open for 60 s, (c) the presence of any contraindication or illness that prevents the participant from undergoing any tests for the evaluation, and (d) not reaching the maximum scores on the Berg Balance Scale and Tinetti Test. Both tests present evidence that supports their ability to predict future falls in the elderly. However, they are not able to detect the early deterioration of balance [20]. The process of selecting the sample is detailed in Figure 1.

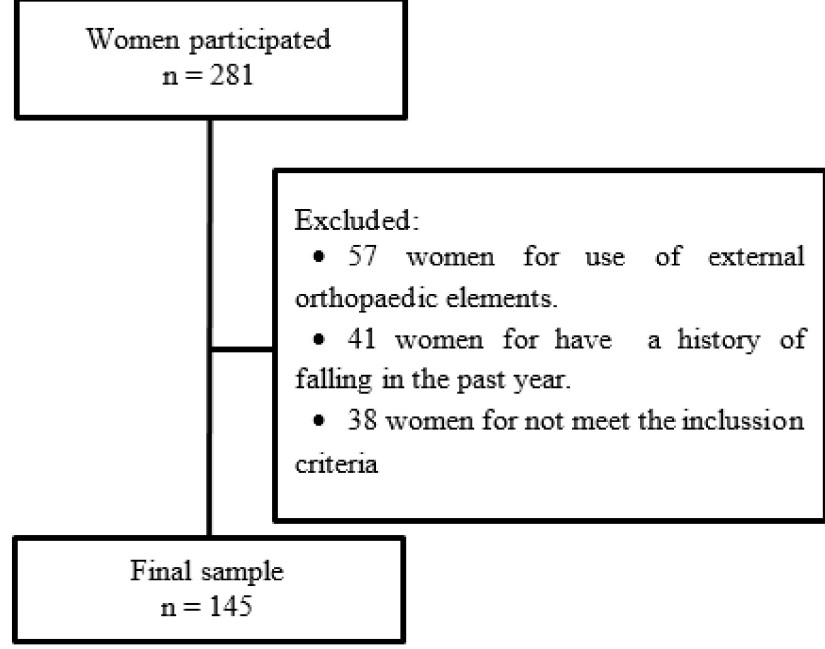

**Figure 1.** Sample CONSORT flow diagram.

All of the women gave written informed consent prior to their participation in the study, in accordance with the Declaration of Helsinki (rev. 2008). Ethical approval was obtained from the Ethics Commission of the Faculty of Education and Sport Sciences of the University of Vigo (Spain; code 3-0406-14).

*2.2. Instruments*

To measure acceleration, we used a triaxial accelerometer ActiGraph GT3X+® (ActiGraph LLC, Pensacola, FL, USA). This accelerometer allows for a time series of acceleration data to be stored in a non-volatile flash memory. The small dimensions of the modules (4.6 × 3.3 × 1.5 cm), attached to its low weight (19 g), accuracy (3 mg/LSB or *Least Significant Bit*) and its range (± 6 units of gravity or G), make these devices a good choice to evaluate the changes in body position in an outpatient environment.

Because accelerometric measurement is only interested in movements that can be attributed to human activity and not the environment under vibration, the signal detected by the accelerometer must pass through a 5 Hz filter before being processed. This threshold makes the device's measurement capability appear limited, especially when it comes to the analysis of populations of persons of advanced age, in which the movements are of a smaller amplitude. Therefore, the selected frequency was 30 Hz. In addition, when working with a low frequency, the noise from the signal is more effectively eliminated.

Accelerometers provide data on the movements along all three axes, namely: axis 1, corresponding to acceleration along the vertical axis; axis 2, along the mediolateral axis; axis 3, along the anterior–posterior axis; and, the root mean square (RMS) of the three axes. The accelerometer measurements were configured for a time frame of 1 s.

*2.3. Procedure*

The tests were performed on barefooted participants wearing socks and comfortable clothing, allowing them to perform the tests comfortably.

The accelerometer was placed directly on the skin at the height of the spinous process of the fourth lumbar vertebra. The locking device was secured with an adjustable belt and hypoallergenic

adhesive tape to ensure that the device did not move independently of the trunk of the subject during the performance of the tests.

First, the participants were asked to travel at a usual pace for a distance of 20 m, divided into two sections of out and back. The beginning and end points of the rotation of the drive were properly marked. The test was repeated three times, each was separated by intervals of 30 s so as to prevent the effect of lower limb muscle fatigue [17]. The results of the three attempts were averaged and were used in the analysis. Then, on the same day as the accelerometric measures, the three balance tests in clinical practice were conducted in the following order: Timed Up and Go, Six-Minute Walk Test and Chair Stand Test. Finally, the body mass index (BMI) and waist perimeter measurements were recorded. The different stages of the procedure are detailed in Figure 2.

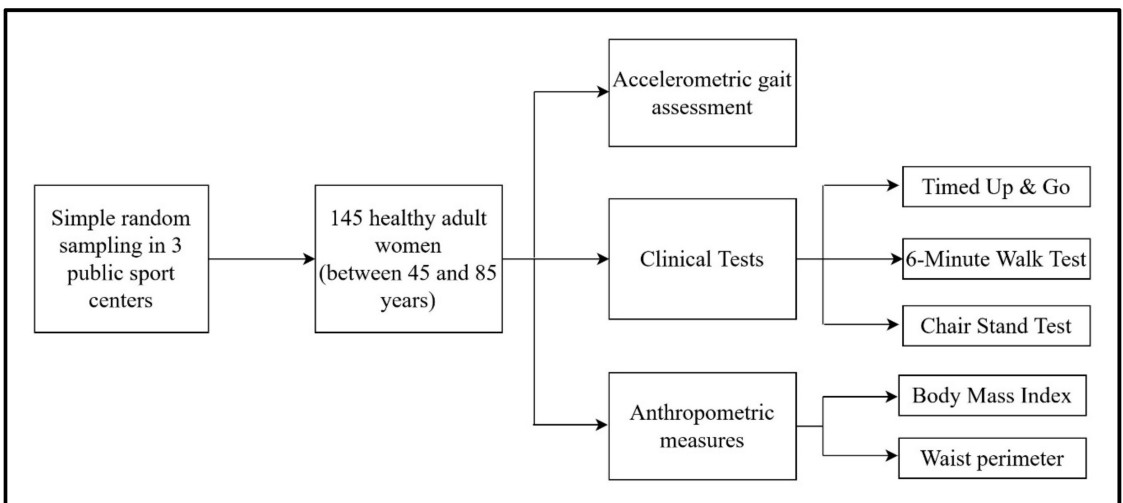

**Figure 2.** Study procedure.

*2.4. Clinical Indicators*

(a)　Timed Up and Go (TUG): a clinical test that evaluates progress in alternating from a sitting to a standing position. It was chosen because it correlates with factors such as the risk of fall or the degree of dependence [21].

(b)　Chair Stand Test (CST): evaluates the strength of the lower limbs through the number of transfers from a sitting to standing position, and vice versa, that the participants were able to perform in 30 s. This test has been correlated with the risk of falling and the steady state [22].

(c)　Six-Minute Walk Test (6MW): used to measure the maximum distance that a person can walk in 6 min. This test is a useful integrated measure of mobility [23].

(d)　Body mass index (BMI) and waist perimeter (WP): both anthropometric variables were chosen for their relationship with the state of physical health in older people [24].

*2.5. Statistical Analysis*

For the analysis of the results, the sample was divided into the following three age groups: G1, between 51 and 60 years (N= 38); G2, between 61 and 70 years (N = 72); and G3, between 71 and 80 years (N = 35). The mean of the three times of the test was used for the analysis. Descriptive statistics used the media as a measure of central tendency and the standard deviation as a measure of dispersion. Analysis of variance (ANOVA) with the Bonferroni correction was used to test for the significance of the differences between the groups. We used linear regression models using the accelerometric means' outcomes (dependent variables) and TUG (independent variable), along with adjustments for age. To evaluate the fit in the linear regression models, the $R^2$ statistic was used. The criteria to evaluate the adjustment values higher than $R^2 > 0.25$ were used as long as they were significant. All

of the calculations were performed using SPSS for Windows version 17.0, and the accepted level of significance was $p < 0.05$.

## 3. Results

There were no significant differences between the groups for weight, BMI or CST. Among groups G2 and G3, there were differences in the 6MW. Only the TUG identified differences between the three age groups (Table 1).

**Table 1.** Descriptive statistics of anthropometric and clinical variables.

| Variables | All (N = 145) | G1 (N = 38) | G2 (N = 72) | G3 (N = 35) |
|---|---|---|---|---|
| Age (years) | 63.8 ± 8.4 | 53.4 ± 5.3 [a**, b**] | 64.2 ± 2.7 [a**, c**] | 74.2 ± 4.6 [b**, c**] |
| Weight (kg) | 65.6 ±10.1 | 63 ± 7.6 | 66.4 ± 11.1 | 66.5 ± 10.1 |
| Height (cm) | 153.9 ± 5.4 | 155.6 ± 5 [b*] | 154 ± 5.5 | 151.8 ± 5.2 [b*] |
| BMI (kg/m$^2$) | 27.6 ± 4.1 | 26 ± 3.3 | 28 ± 4.7 | 28.3 ± 3.2 |
| WP (cm) | 91.5 ± 8.8 | 85.2 ± 6 [a*, b*] | 92.5 ± 8.9 [a*] | 94.5 ± 8.3 [b*] |
| TUG (s) | 6.1 ± 1 | 5.6 ± 0.8 [a*, b**] | 6.1 ± 0.8 [a*, c*] | 6.7 ±1.3 [b**, c*] |
| CST (repetitions) | 20.5 ± 5.3 | 21.5 ± 5.3 | 20.1 ± 5.5 | 20 ± 4.9 |
| 6MW (m) | 588.4 ± 84.5 | 610.2 ± 58.7 [b*] | 594.8 ± 92.7 [c*] | 551.7 ± 81.3 [b*, c*] |

G1—between 51 and 60 years; G2—between 61 and 70 years; G3—between 71 and 80 years; BMI—body mass index; WP—waist perimeter; TUG—Timed Up and Go; CST—Chair Stand Test; 6MW—Six-Minute Walk Test. [a]: G1 vs. G2; [b]: G1 vs. G3; [c]: G2 vs. G3. *: *p* value < 0.05; **: *p* value < 0.001.

There was a reduction in the values of acceleration recorded along all three axes and in the RMS as the age increased (Table 2). This reduction was very significant for the minimum values registered along the vertical and transverse axes, and for the maximum values along the mediolateral axis. Only the maximum values of the vector module demonstrated significant differences among the three age groups. The average duration of the three attempts revealed a difference between G3 and the other two, but did not find a significant difference between G1 and G2.

**Table 2.** Descriptive statistics of the accelerometric gait assessment.

| Variables | | All (N = 145) | G1 (N = 38) | G2 (N = 72) | G3 (N = 35) |
|---|---|---|---|---|---|
| Vertical axis (G) | Maximum | 61.5 ± 16.3 | 68 ± 19.3 [b**] | 61.7 ± 15 | 54.4 ± 12.5 [b**] |
| | Minimum | 4.1 ± 5 | 7 ± 7.2 [a***, b**] | 3.2 ± 3.6 [a***] | 2.9 ± 3.1 [b**] |
| | Mean | 39.3 ± 12.3 | 43.4 ± 14.2 [b*] | 39.9 ± 11.2 [c*] | 33.5 ± 10.3 [b*, c*] |
| Mediolateral axis (G) | Maximum | 48.6 ± 12.9 | 53.9 ± 15.1 [a*, b*] | 47.6 ± 10.94 [a*] | 44.8 ± 12.8 [b*] |
| | Minimum | 5.5 ± 3.7 | 6.5 ± 4.6 [b*] | 5.6 ± 3.5 | 4.2 ± 2.6 [b*] |
| | Mean | 20.7 ± 6.9 | 22.5 ± 8 [b*] | 20.9 ± 6.3 | 18.6 ± 6.1 [b*] |
| Anterior–posterior axis (G) | Maximum | 41.6 ± 11.6 | 45.3 ± 13.3 [b*] | 41.1 ± 10.7 | 38.5 ± 10.4 [b*] |
| | Minimum | 6.9 ± 5.6 | 10 ± 6.9 [a***, b**] | 6.1 ± 4.9 [a***] | 5.1 ± 3.6 [b**] |
| | Mean | 24.2 ± 8.2 | 28 ± 9.2 [a*, b*] | 23.5 ± 7.8 [a*] | 21.5 ± 6.6 [b*] |
| Root mean square (G) | Maximum | 76.6 ± 16.1 | 84.9 ± 19.6 [a*, b***] | 76.4 ± 13.6 [a*, c*] | 68.2 ± 12.1 [b***, c*] |
| | Minimum | 23.7 ± 10.8 | 27.6 ± 13.1 [b*] | 23.2 ± 9.7 | 20.5 ± 9 [b*] |
| | Mean | 54.7 ± 12.7 | 60.5 ± 15.3 [b***] | 54.9 ± 10.9 [c*] | 48 ± 9.7 [b***, c*] |
| Duration (s) | Mean | 16.4 ± 2.5 | 15.6 ± 2.2 [b*] | 16.3 ± 2 [c*] | 17.5 ± 3.1 [b*, c*] |
| Speed (m/s) | Mean | 1.2 ± 0.6 | 1.3 ± 0.5 | 1.2 ± 0.5 | 1.1 ± 0.8 |

G1—between 51 and 60 years; G2—between 61 and 70 years; G3—between 71 and 80 years; [a]: G1 vs. G2; [b]: G1 vs. G3; [c]: G2 vs. G3. *: *p* value < 0.05; **: *p* value < 0.01; ***: *p* value < 0.001.

The correlations of the TUG with the accelerometric variables showed differences according to the age group. In G1, TUG was associated with the maximum value of the anterior–posterior axis ($r = -0.63$, $p < 0.01$). In G2 and G3, TUG was associated with the maximum value of the RMS (G2:

r = −0.52, $p < 0.001$; G3: r = −0.71, $p < 0.001$). All of the groups showed TUG to be correlated with the duration of the accelerometric test (G1: r = 0.71; G2: r = 0.68; G3: r = 0.81; $p < 0.001$ for all of the groups).

The correlations with the 6MW were significant in G1, with maximum values for the mediolateral axis (r = 0.65, $p < 0.01$), anterior–posterior axis (r = 0.72, $p < 0.001$) and RMS (r = 0.54, $p < 0.05$). In G3, the correlations of the CST with the accelerometric variables showed significant correlations with the maximum value of the mediolateral axis (r = 0.69, $p < 0.001$) and the duration of the test (r = −0.68, $p < 0.001$).

The weight was correlated with the minimum value of the RMS in G3 (r = −0.54; $p < 0.05$). The BMI was correlated with the minimum value of the vertical axis (r = −0.65; $p < 0.05$) and the RMS (r = 0.5; $p < 0.05$) in G3. The waist circumference was not correlated with the accelerometric variables in the analysis by group.

Through a model of the linear regression for TUG (Table 3), it was observed that the accelerometric variables that most explained the results of the test were the minimum values of the three axes. To adjust the results of the model for the variable age, the options that provided more information were the minimum value of vertical axis and the maximum values of the mediolateral and anterior–posterior axes and the RMS. The adjustment of the model is reflected in Table 3, and the values of $R^2$ are significant ($p > 0.001$).

**Table 3.** Linear regression models for the Timed Up and Go Test (continuous variable).

| Variables Included | | Crude | | Adjusted for Variable Age | | |
|---|---|---|---|---|---|---|
| | | B | SE | B | SE | $R^2$ |
| Vertical axis | Maximum | −0.02 *** | 0.004 | −0.01 ** | 0.004 | 0.28 *** |
| | Minimum | −0.07 *** | 0.016 | −0.04 ** | 0.015 | 0.27 *** |
| | Mean | −0.02 *** | 0.007 | −0.01 * | 0.006 | 0.27 *** |
| Mediolateral axis | Maximum | −0.03 *** | 0.006 | −0.02 *** | 0.006 | 0.31 *** |
| | Minimum | −0.06 ** | 0.022 | −0.04 | 0.02 | 0.25 *** |
| | Mean | −0.02 | 0.02 | −0.03 | 0.01 | 0.28 *** |
| Anterior–posterior axis | Maximum | −0.04 *** | 0.007 | −0.03 *** | 0.006 | 0.26 *** |
| | Minimum | −0.06 *** | 0.014 | −0.03 * | 0.014 | 0.25 *** |
| | Mean | −0.04 *** | 0.01 | −0.02 * | 0.01 | 0.26 *** |
| Root mean square | Maximum | −0.03 *** | 0.004 | −0.02 *** | 0.005 | 0.37 *** |
| | Minimum | −0.03 *** | 0.008 | −0.02 * | 0.007 | 0.27 *** |
| | Mean | −0.03 *** | 0.006 | −0.02 *** | 0.006 | 0.31 *** |
| Duration | Mean | 0.06 *** | 0.015 | 0.06 *** | 0.002 | 0.53 *** |

B—regression coefficient; SE—standard error; $R^2$: coefficient of determination; *: $p$ value < 0.05; **: $p$ value < 0.01; ***: $p$ value < 0.001.

## 4. Discussion

The aim of this research was to determine whether the accelerometric assessment of gait is able to detect alterations related to natural aging in a population of adult women and older adults. The results suggest that an accelerometer is capable of detecting the differences in gait in women between 51 and 80 years old.

The correlations we found between the clinical variables are consistent with previous research, such as one study that found TUG to be correlated with age [25]. This test was also correlated with the time it took women to complete the test with the accelerometer. Among the clinical trials employed, only TUG displayed significant differences for all of the groups. The correlation of the TUG test with the maximum RMS values is consistent with earlier results, as both parameters have been linked to a risk of falling [26]. Correlations by groups of clinical variables with the axes of space showed a greater dependence on the larger groups with lower limb strength and anthropometry (weight and BMI). These results are consistent with the literature, where the risk of falling is related to a loss of strength and lean mass in the lower limbs, found on numerous occasions [27].

In terms of the individual analysis of each axis, the role played by movement in the sagittal plane is noteworthy. In this sample, there is a reduction of acceleration in this axis as the age increases. Previously, this has been related to exaggerated rolling in the sagittal plane during movement with compensation associated with deterioration [13]. This deterioration is due to the rigidity of the pelvic girdle, and breaks with the physiological premise of the principle of energy economy. Accelerations in the sagittal plane and the RMS have previously been strongly associated with a risk of falling [26]. Consistent with these findings, the maximum values of the mediolateral axis and the minimum values of the RMS are directly correlated with lower limb strength and body composition in the older age group. In relation to movements in the anterior–posterior plane (anterior–posterior axis), the increase in these is also consistent with the absence of pathologies in the sample. The increase in accelerations at this level has been associated with the need to exert a thrust forward from the trunk due to weakness or fatigue in the lower body [28].

The regression model allowed us to identify the values that give more information on the gold standard of clinical balance. The values of $R^2$, although low, are significant. The results identified the maximum values of the RMS and mediolateral axis as being the most important (as well as the analysis of the accelerometric and clinical correlations), in addition to the minimum values of the vertical axis.

Acceleration and speed were reduced as age increased as a result of the increased cadence and reduced stride length [14,19]. This fact is consistent with previous observations that speed is reduced with aging, even in the absence of pathology [15,29]. In addition, the ability to effectively respond to disturbances is maintained, even in G3, as the recorded accelerations were not increased along any of the axes [30].

The results indicate that during aging and in the absence of pathology, in the absence of pathology, speed and acceleration are reduced. The decrease in speed renders the individual more sensitive to falls. However, this occurs to maintain stability despite aging-associated alterations in the neuromotor, muscular and proprioceptive functions [28]. However, if the recorded acceleration does not increase, the sequencing of the gait has not been altered [26,31]. This instrument allows us to identify early changes in spatiotemporal gait parameters. Its use will enable the early diagnosis of signs related to physical and mental deterioration [5,32].

This study has several limitations. First, because of the small sample size, the results have a low generalizability. The second limitation is the absence of males in the sample. The third is the lack of longitudinal data to directly evaluate deterioration in balance.

The results indicate that an accelerometric gait assessment is capable of detecting differences in women between 51 and 80 years old, and allows for an analysis of detailed movements along the three axes of space.

In the study of healthy people where variability in the duration of the tests is small, the analysis of each of the axes can be a source of early diagnosis for the deterioration of balance. An exhaustive analysis of the minimum values of the vertical axis and the maximum values of the mediolateral axis and the vector module allows for the detection of early alterations in the automatic gait pattern.

**Author Contributions:** R.L.-R., J.L.G.-S., V.R.-P. and A.S.-R. contributed to the conceptualization, methodology, software validation, formal analysis, investigation, resources, data curation, writing (original draft) preparation, review and editing and visualization. All authors have read and agreed to the published version of the manuscript

**Funding:** This research received no external funding.

**Conflicts of Interest:** The authors declare no conflict of interest.

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
