# Peer review of "Identification of Body Balance Deterioration of Gait in Women Using Accelerometers"

_sustainability, doi:10.3390/su12031222_

Round 1

Reviewer 1 Report

Journal                      Sustainability/mdpi.com

Manuscript                Sustainability

Title of paper            Identification of balance deterioration in women using accelerometers

General comments

General remarks

The topic of the study – the possibility to identify body balance changes using an accelerometer - is relevant and interesting. The size of the subjects’ group (three groups, total n=145) is sufficient for valid conclusions, the results are obtained with a valid method (triaxial  goniometer), and the measurement procedure is described adequately. Appropriate statistics are used for data analysis. The results are presented in three  tables and two figures designed to show sampling of participants and study procedures. The results are compared to other studies in the Discussion part, and study limitations are provided. The conclusions should be matched with the results of the present study, currently they are mostly general.  The references are in the line of the main topics of the article.

Specific comments:

Title   I recommend rewriting the title in the following way: “ Identification of body balance deterioration during gait in women using accelerometer”.

Abstract   Use description of data changes in three measured groups and results of measurements

Key words   Modify key words according to the article topics (change last two – motor activity and risk factors)

Introduction   Include tests used clinically for human gait description. 

The aim of the study should be formulated in a more specific manner to include the hypothesis of difference between three age groups.

Materials and methods

2.1. Subjects   

Please include the range of height of the participants and distribution into the three groups.

L79-81 Figure 1. Total population of town has no relation to measured participants, please delete these lines.

Table 1 should be here.

Instruments. 

Please provide the details of the accelerometer – producer’s name and country. What are the units of measurement and their accuracy?

The applied method has to be described in a more detailed manner.

Procedures

Ethics description should be in the Participants part.

In which sequence were the tests performed? Please add this information in the beginning of paragraph as well as Fig. 2. Study procedures.

Figure 2. Please add a description of groups here or in Fig. 1. , and Chair Stand Test except of Waist perimeter for Clinical tests.

L 124. Was 30 s rest enough between triple 20-m-walk test? Did you estimate perceived fatigue level in points after each test?

L 130-137. Did participants perform the clinical tests once?

L 150 Please include “... accepted level of significance was p<0.05”.

Results

155-157 Table 1. Please add data for 20 m walking time and, if possible, gait velocity and relation to body height.

In note, add explanation for G1-G3. Same for Table 2.

L 164-165. Please add units for RMS

L174. Describe axis 2 in full words.

L 176. Describe axis 2 in full words.

L183 Add parentheses - (p<0.001)

Table 3 . Add a description of B, Se. R2 in the notes.

Discussion

L 187-189 Please add main results of the study (comparison of data between three groups)

L 231-236 Conclusions should be matched with the results of the present study, at present it is too general. 

Author Response

Dear Editor and Reviewer of the Sustainability Journal:

Thank you very much for your suggestions and contributions to improve the quality of the manuscript. Following your indications, we respond, point by point, to your comments.

Firstly, the English language and style of the manuscript has been reviewed and corrected.

In the text, all the modified or added sentences have been written in red to facilitate the correction by the reviewers.

a) TITLE: I recommend rewriting the title in the following way: “Identification of body balance deterioration during gait in women using accelerometer”.

The authors have modified the title according to your instructions.

b) ABSTRACT: Use description of data changes in three measured groups and results of measurements.

We have expanded the Results section of the ABSTRACT: “There was a reduction in the values of acceleration recorded along all three axes and in Root Mean Square as age increased. This reduction was very significant for the minimum values registered along the vertical and transverse axes and for the maximum values along the mediolateral axis. Only the maximum values of the vector module demonstrated significant differences among the three age groups. A regression model allowed us to identify the values that give more information on the Timed Up & Go Test: the maximum values of Root Mean Square and mediolateral axis.”

c) KEY WORDS: Modify key words according to the article topics (change last two – motor activity and risk factors).

We have changed both key words that indicate by: Falls and Functional Performance.

d) INTRODUCTION: Include tests used clinically for human gait description. The aim of the study should be formulated in a more specific manner to include the hypothesis of difference between three age groups.

In the Introduction section, we have included information about what you recommend: “Research studies have mostly based their results on analyses conducted with force platforms and electronic walkways [4, 5]. This tools provides results based on the behaviour of the center of pressure of the body (based on ground reation forces and spatiotemporal gait parameters). This parameter has been linked to the risk of falling, but it is not a reflection of the overall performance of the body in space [6, 7]. By contrast, in the clinical setting, the use of tests such as the Romberg test, the Timed Up & Go, the Berg Balance Score, the Tinetti Scale, the Functional Reach Test or the BESTest (Balance Evaluation Sistems Test) is frequent. In general, one aspect common to all of them is that the resulting score depends on the qualitative and subjective evaluation of the evaluator [8].”

We have reformulated the objective of the study: “The aim of this research is to determine whether the accelerometric valuation of gait can detect alterations in balance along the aging (through comparing different age groups).”

e) MATERIALS AND METHODS: Please include the range of height of the participants and distribution into the three groups. L79-81 Figure 1. Total population of town has no relation to measured participants, please delete these lines.

We have corrected both aspects.

f) INSTRUMENTS: Please provide the details of the accelerometer – producer’s name and country. What are the units of measurement and their accuracy?

We added that information: “To measure acceleration, we used a triaxial accelerometer ActiGraph GT3X+® (ActiGraph LLC, USA). This accelerometer allows a time series of acceleration data to be stored in a non-volatile flash memory. The small dimensions of the modules (4.6 x 3.3 x 1.5 cm), attached to its low weight (19 g), accuracy (3 mg/LSB) and its range (± 6 units of gravity or G), make of these devices a good choice to evaluate changes in body position in an outpatient environment.”

g) PROCEDURE: Ethics description should be in the Participants part. In which sequence were the tests performed? Please add this information in the beginning of paragraph. Figure 2. Please add Chair Stand Test except of Waist perimeter for Clinical tests.

We have corrected both aspects.

h) L 124. Was 30 s rest enough between triple 20-m-walk test?

No, the gait test was repeated three times, and each one was separated by intervals of 30 seconds to prevent the effect of lower limb muscle fatigue.

i) Did you estimate perceived fatigue level in points after each test?

According to the bibliography consulted and previous articles about gait analysis, separating the attempts for 30 seconds ensures the prevention of fatigue.

j) L 130-137. Did participants perform the clinical tests once?

The complete sequence of tests described was repeated only once. However, the validated protocol of some of these tests involves its performance several times and the recording of the average value of the different attempts (such as Timed Up & Go). This aspect is only indicated in the accelerometric gait test because it is the only one not standardized and previously validated. In the other tests we chose to do a brief description without details of its application to facilitate the reading of the manuscript and because they are very popular tests. In addition, each of the clinical tests are properly referenced.

k) L 150 Please include “... accepted level of significance was p < 0.05”.

We have included that sentence.

l) RESULTS: 155-157 Table 1. Please add data for 20 m walking time and, if possible, gait velocity. In note, add explanation for G1-G3. Same for Table 2. L 164-165. Please add units for RMS.

Both variables have been included in Table 2 (because they are variables extracted from the accelerometric gait test).

In the notes of Tables 1 and 2, we have added the explanation of G1, G2 and G3.

We have asses the unit for RMS: unit of gravity (G).

m) RESULTS: L174. Describe axis 2 in full words. L 176. Describe axis 2 in full words. L183 Add parentheses - (p < 0.001).

We have corrected the three errors.

n) RESULTS: Table 3. Add a description of B, Se. R2 in the notes.

We added a note with that description.

o) DISCUSSION: L 187-189 Please add main results of the study (comparison of data between three groups). L 231-236 Conclusions should be matched with the results of the present study, at present it is too general.

We have rewritten the first paragraph of the Discussion section and modified the conclusions.

Once again, thank you very much for the time spent and the interest shown in this work; as well as in the positive evaluations you have given of it.

Receive a warm greeting,

The authors.

Reviewer 2 Report

Dear editors,

I have reviewed the downloaded manuscript for a few days. This study is perfectly worked. I have read other studies of Dr. Romo and have a correct style of performing the treatment statistically and methodologically, the study is correct.

I approve the publication of the manuscript. I have not found errors in it.

Author Response

Dear Editor and Reviewer of the Sustainability Journal:

Firstly, the English language and style of the manuscript has been reviewed and corrected.

In the text, all the modified or added sentences have been written in red to facilitate the correction by the reviewers.

Thank you very much for the positive evaluation of our study.

Although you do not ask us to correct any details of the study, we have corrected it following the advice of the other reviewer. The authors hope that this study will be even more enjoyable now.

Thank you very much for the time spent and the interest shown in this work; as well as in the positive evaluations you have given of it.

Receive a warm greeting,

The authors.
